# MULTI-AGENT VERIFICATION: SCALING TEST-TIME COMPUTE WITH MULTIPLE VERIFIERS (ABRIDGED)*

**Shalev Lifshitz**
ArdaLabs.AI
shalev@ardalabs.ai

**Sheila A. McIlraith**
University of Toronto
Vector Institute for AI
sheila@cs.toronto.edu

**Yilun Du**
Harvard University
ydu@seas.harvard.edu

## ABSTRACT

By utilizing more computational resources at test-time, large language models (LLMs) can improve without additional training. One common strategy uses *verifiers* to evaluate candidate outputs. In this work, we propose a novel scaling dimension for test-time compute: *scaling the number of verifiers*. We introduce Multi-Agent Verification (MAV) as a test-time compute paradigm that combines multiple verifiers to improve performance. We propose using Aspect Verifiers (AVs), off-the-shelf LLMs prompted to verify different aspects of outputs, as one possible choice for the verifiers in a MAV system. AVs are a convenient building block for MAV since they can be easily combined without additional training. Moreover, we introduce BoN-MAV, a simple multi-agent verification algorithm that combines best-of-$n$ sampling with multiple verifiers. BoN-MAV demonstrates stronger scaling patterns than self-consistency and reward model verification, and we demonstrate both weak-to-strong generalization, where combining weak verifiers improves even stronger LLMs, and self-improvement, where the same base model is used to both generate and verify outputs. Our results establish scaling the number of verifiers as a promising new dimension for improving language model performance at test-time.

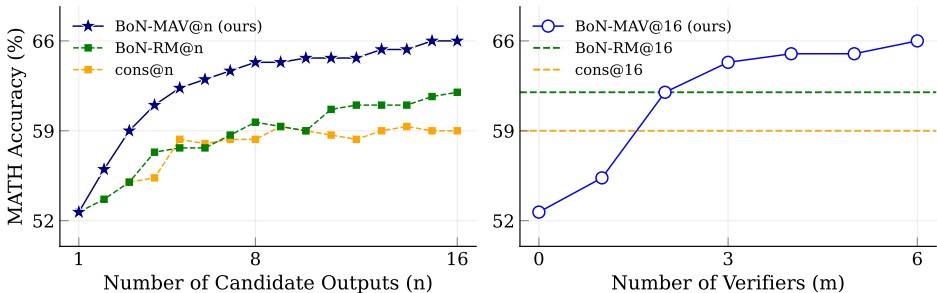

Figure 1: **Scaling test-time compute along two dimensions.** *Left:* Increasing the number of candidate outputs ($n$) and comparing three test-time methods: best-of-$n$ with multi-agent verification (BoN-MAV@n), best-of-$n$ with reward model verification (BoN-RM@n), and self-consistency (cons@n). *Right:* Increasing the number of verifiers ($m$) when selecting between $n = 16$ candidate outputs (BoN-MAV@16) surpasses the performance of reward model verification (BoN-RM@16) and self-consistency (cons@16). All candidate outputs are sampled from Gemini-1.5-Flash on the MATH benchmark (Hendrycks et al., 2021).

## 1 INTRODUCTION

Scaling the size of large language models (LLMs) and their training datasets has driven remarkable progress in artificial intelligence (Brown et al., 2020; Chowdhery et al., 2023; Hoffmann et al., 2022). However, the growing cost of scaling model size and obtaining unseen high-quality pretraining data has sparked growing interest in methods that improve LLM performance without simply scaling parameters or data. Among these, a promising new direction has emerged: *scaling test-time compute*, where models spend more computational resources during inference—much like humans spend more time thinking through harder problems.

---

*Full version of the paper is available at https://arxiv.org/abs/2502.20379, and see https://ardalabs.ai/MultiAgentVerification for the paper webpage.

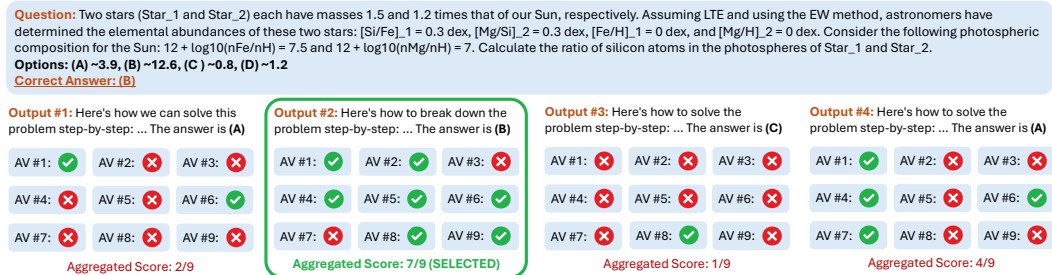

Figure 2: **Illustration of the BoN-MAV algorithm.** BoN-MAV combines best-of-$n$ sampling with multi-agent verification: First, $n$ candidate outputs are sampled from a generator LLM. Then, each output is evaluated by a set of Aspect Verifiers (AVs) that produce binary approvals. Finally, the candidate with the most approvals is selected as the final answer. See Section 2.3 for algorithm details.

A common strategy for scaling test-time compute is *best-of-n sampling* (Stiennon et al., 2020; Cobbe et al., 2021; Nakano et al., 2021), where $n$ candidate outputs are sampled from a *generator* LLM and a *verifier* model scores each candidate output based on its quality or correctness. The highest-scoring output is then selected. Under this strategy, the amount of test-time compute can be scaled up by increasing the number of sampled outputs. However, in this work, we propose a new orthogonal scaling dimension: *scaling the number of verifiers*. We introduce **Multi-Agent Verification (MAV)**, a test-time compute paradigm that combines multiple verifiers to improve performance.

Typically, verifiers are implemented as reward models which are trained using reinforcement learning from human feedback (Christiano et al., 2017; Stiennon et al., 2020; Ouyang et al., 2022; Bai et al., 2022a). However, relying on reward models as verifiers introduces two crucial limitations for multi-agent verification: (1) each reward model has to be trained on expensive curated preference data, and (2) there is no straightforward way to combine scores generated by heterogeneous reward models trained on different datasets (they produce uncalibrated scores). These limitations make reward models poorly suited for multi-agent verification and restrict our ability to simply scale up the number and type of verifiers at test-time.

To address these limitations and enable scalable multi-agent verification, we propose using **Aspect Verifiers (AVs)** — off-the-shelf LLMs prompted to verify specific aspects of candidate outputs through binary True/False approvals. This approach is motivated by the observation that internet data contains abundant examples of humans providing binary evaluations with feedback (e.g., educational assessments, academic peer reviews, online forums, and automated code tests), which suggests that language models may be naturally suited for binary verification. Unlike reward models, AVs do not require additional training since producing binary approvals falls naturally within the training distribution of their base LLMs, and their binary outputs can be easily combined across multiple models through simple voting mechanisms. Thus, the number and type of aspect verifiers can be easily scaled up without additional training. We note that aspect verifiers are just one possible implementation choice for the verifiers in a MAV system, which address the two key limitations of typical reward model verifiers.

To investigate scaling multi-agent verification, we introduce BoN-MAV as a specific algorithm which combines best-of-$n$ sampling with aspect verifiers. This is one implementation of a MAV algorithm, combining traditional best-of-$n$ sampling with multiple verifiers. Given an input, BoN-MAV (1) samples $n$ outputs from a generator LLM, (2) collects binary approvals from a set of $m$ aspect verifiers, and (3) selects the output with the most approvals. We investigate scaling test-time compute with this approach along two orthogonal dimensions: the traditional dimension of increasing the number of sampled candidate outputs $n$, and our novel test-time scaling dimension of increasing the number of verifiers $m$. We find that using multiple diverse verifiers to select between candidate outputs is an effective strategy, and that performance improves as we use more verifiers. We also demonstrate *weak-to-strong generalization* (Burns et al., 2023), whereby combining many small aspect verifiers can improve the performance of even stronger generator LLMs, and we show that BoN-MAV enables *self-improvement* by using the same base LLM for both the generator and set of aspect verifiers.

Overall, our paper makes the following contributions:

**(1)** We introduce Multi-Agent Verification (MAV) as a new test-time paradigm that combines multiple verifiers at test-time, opening a novel scaling dimension: *scaling the number of verifiers*.

**(2)** We propose Aspect Verifiers (AVs), off-the-shelf LLMs which require no additional training and naturally support combining verification signals from multiple heterogeneous verifiers using voting mechanisms.

**(3)** We demonstrate that BoN-MAV, a simple multi-agent verification algorithm which combines best-of-$n$ with aspect verifiers, improves the performance of various generator LLMs as we scale up the number and type of aspect verifiers.

## 2 MULTI-AGENT VERIFICATION

Multi-Agent Verification (MAV) is a test-time compute paradigm where multiple verifiers are combined to evaluate outputs from a generator LLM. To implement a MAV algorithm, we must address two questions: **(1)** What type of verifiers can be easily combined and scaled up in number without additional training? **(2)** How should we aggregate verification signals from multiple verifiers? In this section, we propose answers to these questions and describe one simple implementation of a multi-agent verification algorithm called BoN-MAV. We discuss future directions for alternative multi-agent verification algorithms in Appendix E.

In Section 2.1, we propose Aspect Verifiers (AVs) as a convenient building block for MAV, since they require no additional training and naturally support combining multiple verification signals. In Section 2.2, we describe our approach to aggregating signals across multiple AVs. In Section 2.3, we outline the BoN-MAV algorithm, which combines best-of-$n$ sampling with aspect verifiers. Section 2.4 proposes verifier engineering to select relevant verifiers for specific domains or tasks.

### 2.1 ASPECT VERIFIERS

In the context of test-time computation with LLMs, a *verifier* typically refers to a model that evaluates the quality or correctness of an output sampled from a generator LLM. Here, we ask: *What type of verifiers can be easily combined and scaled up in number without additional training?*

Prior works have largely focused on using neural reward models as verifiers (Stiennon et al., 2020; Cobbe et al., 2021; Snell et al., 2024). However, these models present key challenges for scaling multi-agent verification. First, each reward model requires training on expensive curated preference data to produce reliable reward scores (Stiennon et al., 2020). Second, while ensembles of homogeneous reward models (identical model initializations trained on the same data but with different random seeds) have been proposed as a way to mitigate overoptimization (Coste et al., 2023; Eisenstein et al., 2023; Gao et al., 2023a), there is no straightforward way to combine scores from heterogeneous reward models trained on different datasets. This second limitation arises because scores from different reward models are uncalibrated——they operate on different numerical scales based on their distinct training setups. We wish to simply scale up the number and type of verifiers without additional training.

We propose Aspect Verifiers (AVs) as one possible implementation choice for the verifiers in a MAV system, which address the two mentioned limitations of typical reward model verifiers. AVs are off-the-shelf LLMs prompted to evaluate specific aspects of candidate outputs and produce binary True/False approvals. Unlike reward models, they require no additional training since binary evaluation is a natural task for LLMs (the internet contains abundant examples of humans providing binary approvals with explanation, such as educational assessments, academic peer reviews, or online forums), and their binary approvals can be easily combined through voting mechanisms, even when AVs are based on completely different models or training data. Using aspect verifiers, we can easily scale up the number and type of verifiers which may be based on different LLMs, training algorithms, architectures, data, or prompts.

Aspect Verifiers can be configured along three axes: (1) The base LLM (e.g., GPT-4o-mini or Gemini-1.5-Flash), (2) The aspects to verify (e.g., mathematical correctness, logical soundness, factuality, etc.), (3) The verification strategy (e.g., direct approval, going over the output step-by-step, rephrasing, checking edge cases, etc.). By varying these three axes, we can create a diverse set of aspect verifiers with differing capabilities. Figure 5 illustrates how multiple aspect verifiers can evaluate a single candidate output, and Appendix G shows the full list used in this work.

## 2.2 Combining Aspect Verifiers

With aspect verifiers as our building block, we ask: *How can we effectively aggregate verification signals across multiple AVs?* We take the simplest possible approach in our experiments: each binary True/False approval is a single vote, and the aggregated score for a candidate output is the sum of the positive votes from all AVs. That is, the aggregated verification score is the sum of the individual binary scores from each verifier:

$$\text{AggScore}(o^{(i)}) = \frac{1}{|\mathcal{M}|} \sum_{v \in \mathcal{M}} \text{BinaryScore}_v(o^{(i)}), \tag{1}$$

where $o^{(i)} \in \mathcal{O}$ is the $i$th candidate output from the set of sampled outputs $\mathcal{O}$, $\mathcal{M}$ is the set of aspect verifiers, and $\text{BinaryScore}_v : \mathcal{O} \to \{0, 1\}$ maps a candidate output from $\mathcal{O}$ to the binary approval produced by verifier $v \in \mathcal{M}$ for that output. This voting strategy gives equal weight to all verifiers in the final aggregated score, and it proves remarkably effective in our experiments (see Section 3.1). However, future works could investigate more sophisticated aggregation strategies such as grouping verifiers by aspect and then voting across aspects, or having aspect verifiers debate with each other (Du et al., 2023) before producing an approval. We discuss these and other potential directions for future work in Appendix E.

## 2.3 BoN-MAV

Best-of-$n$ (BoN) sampling is a test-time optimization technique (Stiennon et al., 2020; Cobbe et al., 2021; Nakano et al., 2021) where $n$ candidate outputs are sampled from a generator LLM, each candidate is scored by a verifier model, and the highest-scoring output is selected. We introduce BoN-MAV as a simple multi-agent verification algorithm that combines best-of-$n$ sampling with aspect verifiers. It uses the simple aggregation strategy from Equation 1 and consists of three steps: (1) sampling $n$ candidate outputs from a generator LLM, (2) collecting binary approvals from a set of $m$ aspect verifiers, and (3) selecting the output with the most approvals. That is, $\hat{i} = \arg \max_{0 < i < n} \left( \text{AggScore}(o^{(i)}) \right)$ where $o^{(i)}$ is the $i$th candidate output, $n$ is the total number of sampled candidate outputs, and $\hat{i}$ is the index of the output with the highest aggregated score (the selected output). Figure 2 illustrates how BoN-MAV can select between a set of candidates.

Using BoN-MAV, we can increase test-time computation by sampling more candidate outputs (increasing $n$) and by querying more verifiers (increasing $m = |\mathcal{M}|$), where test-time computation can be easily parallelized during generation as well as verification. In addition, BoN-MAV represents just one specific approach to multi-agent verification, and more nuanced aggregation algorithms or alternatives to aspect verifiers could further enhance performance (see Appendix E for discussion).

## 2.4 Verifier Engineering

Using aspect verifiers, we can create a diverse pool of verifiers with different capabilities. However, not all verifiers are equally relevant for every domain. Thus, we propose *verifier engineering* as a process to select a subset of verifiers most effective for a particular domain (similar to prompt engineering, where prompts are engineered for specific domains or tasks). We engineer domain-specific sets of verifiers by first creating a diverse initial set $\mathcal{M}$ and then selecting the subset $\mathcal{M}^d \subseteq \mathcal{M}$ which contains the most relevant verifiers for domain $d$. Specifically, for each domain $d$, we select the subset $\mathcal{M}^d \subseteq \mathcal{M}$ which maximizes the average performance across all generator LLMs evaluated on a validation set. Our current approach keeps the engineered set of verifiers fixed for all questions in a domain, but future works could explore dynamically customizing verifiers for particular questions, as we discuss in Appendix E.

## 3 Experiments

In our experiments, we investigate scaling test-time compute along two orthogonal dimensions: the traditional dimension of increasing the number of sampled candidate outputs $n$, and our novel test-time scaling dimension of increasing the number of verifiers $m$. We aim to address the following questions: **(1)** How well does multi-agent verification improve performance across diverse domains and various generator LLMs? **(2)** Can multi-agent verification facilitate weak-to-strong generalization and self-improvement? **(3)** How important is engineering a domain-specific set of verifiers and what are the important design choices? To address these questions, we evaluate the BoN-MAV algorithm described in Section 2 on four domains: MATH dataset (Hendrycks et al., 2021), MMLU-Pro (Wang et al., 2024c), GPQA (diamond) (Rein et al., 2023), and HumanEval (Chen et al., 2021) (see Appendix G.1 for details).

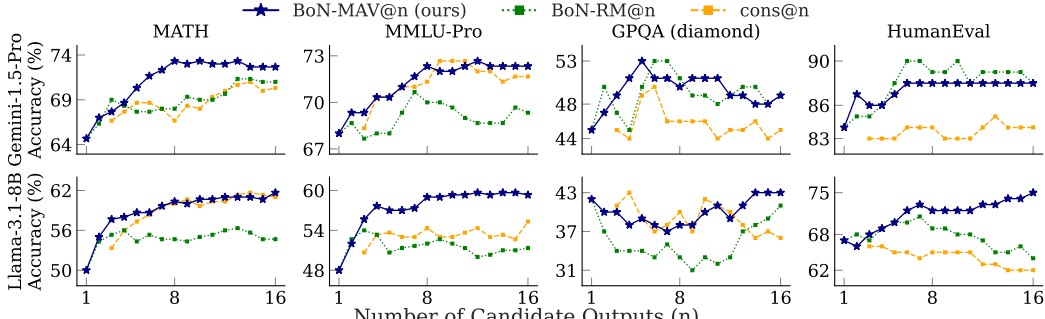

Figure 3: **Scaling the number of candidate outputs.** Performance (accuracy %) of test-time compute methods as we increase the number of sampled candidate outputs ($n$), shown for two generator LLMs across all evaluation domains (see Figure 6 for more models).

## 3.1 MAV ENABLES SCALING ALONG TWO DIMENSIONS

**Baselines.** We compare Best-of-$n$ sampling with Multi-Agent Verification (BoN-MAV) against two established test-time compute methods: (1) best-of-$n$ sampling with reward model verification (Stiennon et al., 2020; Cobbe et al., 2021; Nakano et al., 2021), where we use a trained neural reward model as the external verifier to select the highest-scoring candidate output, and (2) self-consistency (Wang et al., 2022; Li et al., 2022b; Thoppilan et al., 2022; Lewkowycz et al., 2022), which selects the most common answer from the set of candidates outputs. For reward model verification, we use the current top-performing open-source 8B reward model on RewardBench (Lambert et al., 2024) (see Appendix G.5 for more details). Moreover, we implement the verifier engineering method described in Section 2.4 (see Appendix G.2 for more details).

**Quantitative Results.** We evaluate BoN-MAV across four domains using eight generator LLMs (four closed-source and four open-source). For each model, we sample $n = 16$ candidate outputs per question and compare between best-of-$n$ with Multi-Agent Verification (BoN-MAV), best-of-$n$ with reward model verification (BoN-RM), and self-consistency (cons). As shown in Table 1, BoN-MAV outperforms self-consistency in nearly all cases, and outperforms reward model verification on MATH and MMLU-Pro, while achieving comparable results on GPQA (diamond) and HumanEval.

**Qualitative Examples.** Figure 5 illustrates how multiple aspect verifiers can be used to evaluate a single candidate output. The first aspect verifier uses direct yes/no approval without step-by-step thinking and incorrectly approves the output while additional aspect verifiers, using the same base model but with more thorough verification strategies, successfully identify the error. Additional examples are provided in Appendix I. Note that for the purposes of illustration, we visualize slightly different sets of verifiers than the final domain-specific sets used in our experiments.

**Scaling the Number of Candidate Outputs.** In Figure 3, we show the scaling patterns for generator LLMs as we increase the number of sampled candidate outputs ($n$). Matching the results in Table 1, BoN-MAV demonstrates more effective scaling patterns than self-consistency across all domains, and stronger scaling than reward model verification on MATH and MMLU-Pro while achieving comparable scaling patterns on GPQA (diamond) and HumanEval.

**Scaling the Number of Verifiers.** Multi-Agent Verification introduces a powerful new dimension for scaling test-time compute: *scaling the number of verifiers*. In Figure 4, we show how accuracy tends to improve as we increase the number of verifiers $m$ from zero verifiers up to the full domain-specific subset $\mathcal{M}^d$. For each value of $m \in \{0, 1, 2, ..., |\mathcal{M}^d|\}$, we plot the average accuracy across all possible combinations of $m$ verifiers drawn from $\mathcal{M}^d$, with the shaded regions indicating the spread of observed values. Our results demonstrate that scaling verifier count is a promising new dimension for improving model performance at test-time. In most cases, accuracy improves as we add verifiers, with performance gains of up to 10% for large LLMs and up to 20% for small ones. Notably, performance gains persist even when strong generator LLMs (Gemini-1.5-Pro, GPT-4o) are verified by combinations of our weaker verifiers (Gemini-1.5-Flash, GPT-4o-mini), supporting our findings about weak-to-strong generalization in Section 3.2. However, the magnitude and pattern of improvement varies and, in some cases, accuracy initially decreases before improving with additional verifiers. We expect better-engineered verifiers to unlock even stronger scaling patterns.

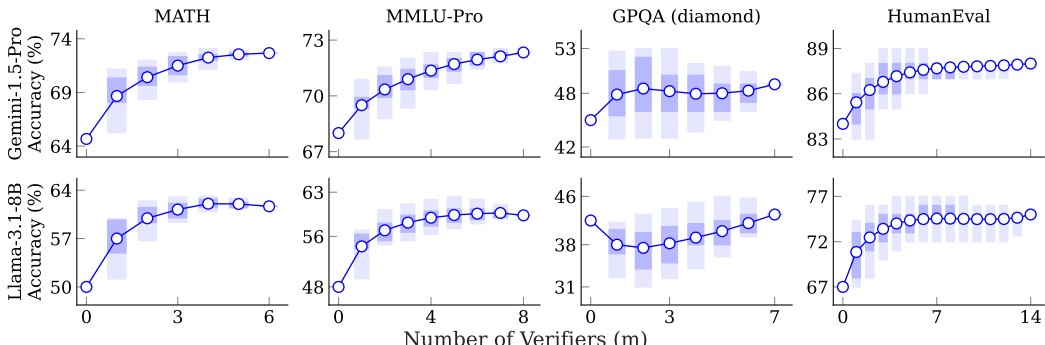

Figure 4: **Scaling the number of verifiers.** Performance (accuracy %) of BoN-MAV as we increase the number of verifiers ($m$) up to domain-specific subsets $\mathcal{M}^d$ (detailed in Section 3.1). For each $m$, we plot the performance of BoN-MAV averaged across all possible combinations of $m$ verifiers drawn from $\mathcal{M}^d$, with shading indicating the spread of observed values — dark blue shows the 25-75th percentile range (middle 50%) while light blue shows the 5-95th percentile range (90% of outcomes). The leftmost point ($m = 0$) represents pass@1 accuracy without verification while the rightmost point ($m = |\mathcal{M}^d|$) uses all verifiers in $\mathcal{M}^d$. See Figure 7 in appendix for more models.

### 3.2 MAV ENABLES WEAK-TO-STRONG GENERALIZATION AND SELF-IMPROVEMENT

**Weak-to-Strong Generalization.** Prior work has shown that weak supervisors can improve the performance of strong pretrained models (Burns et al., 2023). Here, we show that multi-agent verification can be used to enhance the performance of strong generator LLMs by combining weaker verifiers. As shown in Table 2, our strongest generators (Gemini-1.5-Pro and GPT-4o) show substantial improvements over their base pass@1 accuracy when using verifiers based on weaker models (Gemini-1.5-Flash and GPT-4o-mini), and Figure 4 shows how the performance of Gemini-1.5-Pro and GPT-4o changes as we scale the number of verifiers. These results suggest that the diverse perspectives of multiple smaller, computationally cheaper models can collectively produce a verification signal robust enough to improve even state-of-the-art generators.

**Self-Improvement.** Multi-agent verification can also enable models to improve their own performance through self-verification. To demonstrate, we configure BoN-MAV to use the same base LLM for both generation and verification. That is, we sample outputs from a generator LLM (Gemini-1.5-Flash or GPT-4o-mini) and create multiple aspect verifiers derived from the same LLM. Following the verifier engineering procedure from Section 3.1, we select the best subset of self-verifiers based on validation performance. As shown in Table 2, this self-verification approach yields substantial improvements over base pass@1 accuracy across all domains except HumanEval. For instance, GPT-4o-mini shows particularly strong self-improvement on MATH (+7%) and GPQA (+8%).

## 4 CONCLUSION

We have introduced Multi-Agent Verification (MAV), a test-time compute paradigm that combines multiple verifiers to improve performance. MAV enables test-time scaling along two orthogonal dimensions: (1) the traditional dimension of increasing the number of candidate outputs sampled from a *generator* LLM, and (2) our novel test-time scaling dimension of increasing the number of *verifiers* evaluating each output. We propose Aspect Verifiers (AVs) as one possible implementation choice for the verifiers in a MAV system. AVs are off-the-shelf LLMs that require no additional training and naturally support combining verification signals from models based on different LLMs, training algorithms, architectures, data, or prompts. We introduce BoN-MAV as a simple multi-agent verification algorithm and our results indicate that increasing the number of diverse verifiers is a promising dimension for scaling test-time compute. Specifically, we demonstrate improved test-time performance across multiple domains and generator LLMs, weak-to-strong generalization, and self-improvement. BoN-MAV represents just one approach to multi-agent verification and we expect better-engineered verifiers and more nuanced aggregation strategies to unlock even stronger scaling patterns. We discuss the limitations of our approach and potential directions for future research in Appendix E. We hope that our work inspires future research into multi-agent verification algorithms and further exploration of scaling the number of verifiers during test-time.

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

## A    ACKNOWLEDGMENTS

We thank Romi Lifshitz, Andrew Li, Parand Alamdari, Claas Voelcker, Lev McKinney, Toryn Klassen, and David Glukhov for their helpful comments. We thank ArdaLabs for providing funding and compute resources for this project (https://ardalabs.ai/). The second author gratefully acknowledges funding from the Natural Sciences and Engineering Research Council of Canada (NSERC) and the Canada CIFAR AI Chairs Program. Resources used in preparing this research were provided, in part, by the Province of Ontario, the Government of Canada through CIFAR, and companies sponsoring the Vector Institute for Artificial Intelligence (https://vectorinstitute.ai/partnerships/).

## B    VISUALIZATION AND TABLES

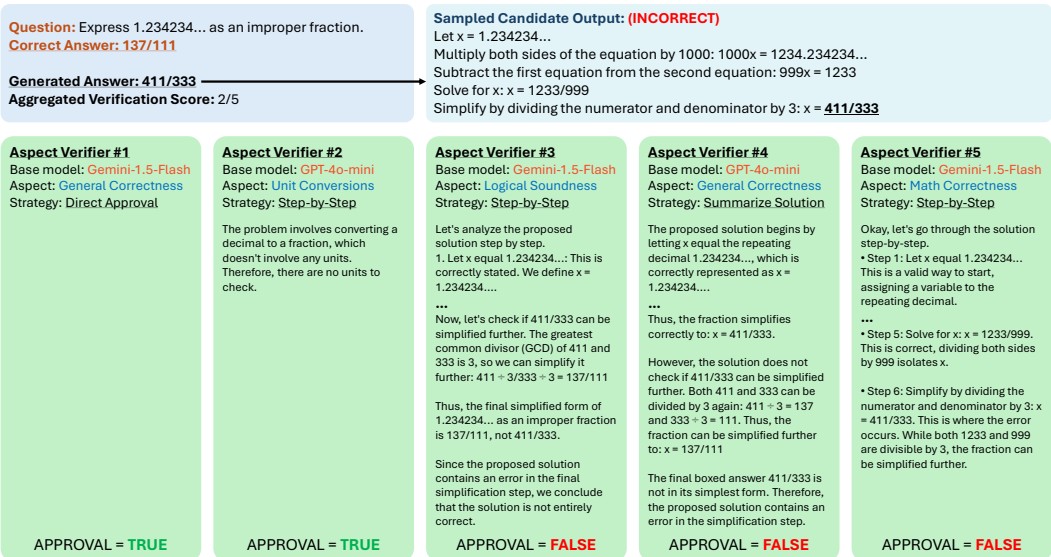

Figure 5: **Multi-agent verification for a single solution.** An illustration of how combining multiple aspect verifiers can produce a more robust verification signal. Five different aspect verifiers evaluate an incorrect MATH (Hendrycks et al., 2021) solution sampled from Gemini-1.5-Pro. The verifiers vary across three dimensions: base models (Gemini-1.5-Flash, GPT-4o-mini), aspects to verify (e.g., general correctness, mathematical correctness, unit conversions), and verification strategies (e.g., direct yes/no approval, step-by-step verification, summarization). Two verifiers miss the error: one using direct approval without step-by-step thinking, and another tasked with checking unit conversions (the problem contains no units to convert, so the verifier finds no errors and incorrectly approves the solution). The remaining three verifiers each identify the mistake through careful analysis. This demonstrates how combining diverse verification methods can produce a robust signal despite individual verifier failures, as the majority correctly identify the error.

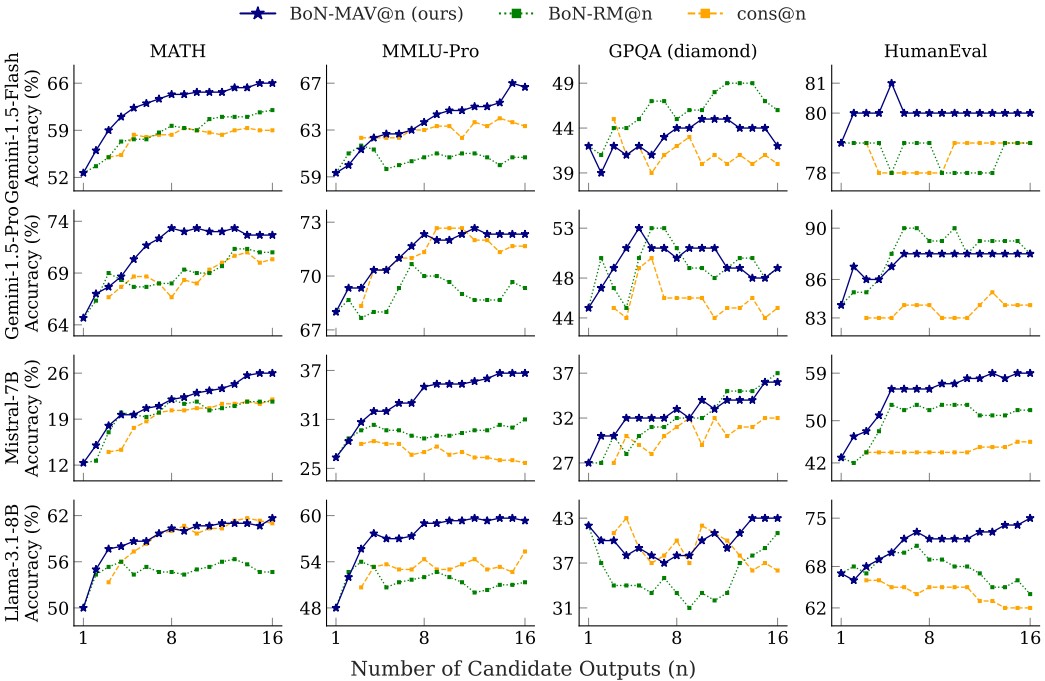

Figure 6: **Scaling the number of candidate outputs.** Performance (accuracy %) of test-time compute methods as we increase the number of sampled candidate outputs ($n$), shown for four generator LLMs (Gemini-1.5-Flash, Gemini-1.5-Pro, Mistral-7B, and Llama-3.1-8B) across all evaluation domains.

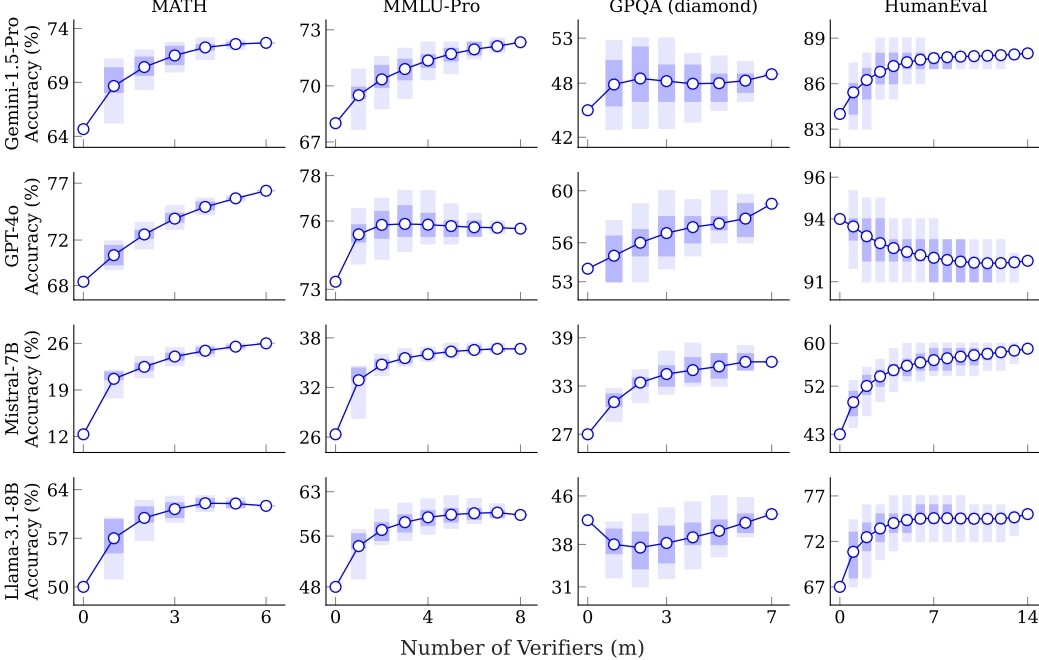

Figure 7: **Scaling the number of verifiers.** Performance (accuracy %) of BoN-MAV as we increase the number of verifiers ($m$) up to domain-specific subsets $\mathcal{M}^d$ (detailed in Section 3.1). For each $m$, we plot the performance of BoN-MAV averaged across all possible combinations of $m$ verifiers drawn from $\mathcal{M}^d$, with shading indicating the spread of observed values — dark blue shows the 25-75th percentile range (middle 50%) while light blue shows the 5-95th percentile range (90% of outcomes). The leftmost point ($m = 0$) represents pass@1 accuracy without verification while the rightmost point ($m = |\mathcal{M}^d|$) uses all verifiers in $\mathcal{M}^d$.

| Generator LLM | MATH B-MAV | Cons | RM | MMLU-Pro B-MAV | Cons | RM | GPQA (diamond) B-MAV | Cons | RM | HumanEval B-MAV | Cons | RM |
|---|---|---|---|---|---|---|---|---|---|---|---|---|
| **Gemini-1.5-Flash** | **66.0** | 59.0 | 61.7 | **66.7** | 63.3 | 60.7 | 42.0 | 40.0 | **46.0** | **80.0** | 79.0 | 79.0 |
| **Gemini-1.5-Pro** | **72.7** | 70.3 | 71.0 | **72.3** | 71.7 | 69.3 | **49.0** | 45.0 | **49.0** | **88.0** | 84.0 | **88.0** |
| **GPT-4o-mini** | 73.0 | **74.7** | 72.3 | **67.0** | 63.7 | 62.7 | **50.0** | 48.0 | 44.0 | 84.0 | **87.0** | 85.0 |
| **GPT-4o** | 76.3 | 77.3 | **80.7** | 75.7 | **76.3** | 72.7 | **59.0** | **59.0** | 58.0 | 92.0 | **95.0** | 92.0 |
| **Mistral-7B** | **26.0** | 22.0 | 21.7 | **36.7** | 25.7 | 31.0 | 36.0 | 32.0 | **37.0** | **59.0** | 46.0 | 52.0 |
| **Llama-3.1-8B** | **61.7** | 61.0 | 54.7 | **59.3** | 55.3 | 51.3 | **43.0** | 36.0 | 41.0 | **75.0** | 62.0 | 64.0 |
| **Gemma-2-9B** | **58.7** | 51.7 | 55.0 | **57.7** | 54.3 | 54.7 | 34.0 | 36.0 | **38.0** | 32.0 | 25.0 | **51.0** |
| **Gemma-2-27B** | **62.3** | 55.7 | 59.3 | **62.0** | 58.3 | 60.0 | **41.0** | 40.0 | **41.0** | **76.0** | 66.0 | **76.0** |

Table 1: **Best-of-$n$ with Multi-Agent Verification (BoN-MAV) across models and domains.** Performance (accuracy %) comparison of three test-time verification methods using $n = 16$ candidate outputs: the BoN-MAV algorithm (labeled as B-MAV in the table), reward model verification (RM), and self-consistency (Cons). Results are shown for eight generator LLMs across four domains, with BoN-MAV on each domain using the domain-specific aspect verifier subset $\mathcal{M}^d$. BoN-MAV outperforms self-consistency in nearly all cases, and generally outperforms RM except on GPQA and HumanEval, where BoN-MAV and RM are comparable.

| | Generator LLM | MATH B-MAV | pass@1 | MMLU-Pro B-MAV | pass@1 | GPQA (diamond) B-MAV | pass@1 | HumanEval B-MAV | pass@1 |
|---|---|---|---|---|---|---|---|---|---|
| **WtoS** | **Gemini-1.5-Pro** | **72.7** | 64.7 | **72.3** | 68.0 | **49.0** | 45.0 | **88.0** | 84.0 |
| | **GPT-4o** | **76.3** | 68.3 | **75.7** | 73.3 | **59.0** | 54.0 | 92.0 | **94.0** |
| **SI** | **Gemini-1.5-Flash** | **59.0** | 52.7 | **64.0** | 59.3 | **43.0** | 42.0 | 78.0 | **79.0** |
| | **GPT-4o-mini** | **76.0** | 69.0 | **65.7** | 62.3 | **46.0** | 38.0 | **86.0** | 86.0 |

Table 2: **Weak-to-strong generalization and self-improvement.** Performance (accuracy %) using the BoN-MAV algorithm (labeled as B-MAV in the table) compared to base pass@1 accuracy. For weak-to-strong generalization (top, "WtoS"), we use aspect verifiers based on weaker models (Gemini-1.5-Flash and GPT-4o-mini) to improve stronger generator LLMs. For self-improvement (bottom, "SI"), we use aspect verifiers based on the same model as the generator. BoN-MAV improves performance in nearly all cases.

| | Ablation 1: Verifier-Engineering | | | | Ablation 2: Verifier Diversity | | | |
|---|---|---|---|---|---|---|---|---|
| | MMLU-Pro | | GPQA (diamond) | | MMLU-Pro | | GPQA (diamond) | |
| **Generator LLM** | Eng | All | Eng | All | Diverse | Same | Diverse | Same |
| **Gemini-1.5-Flash** | **66.7** | 65.7 | **42.0** | 41.0 | **66.7** | 66.3 | **42.0** | 39.0 |
| **Gemini-1.5-Pro** | **72.3** | 70.3 | **49.0** | **49.0** | **72.3** | 71.0 | 49.0 | **55.0** |
| **GPT-4o-mini** | **67.0** | 65.3 | **50.0** | 49.0 | **67.0** | 64.7 | **50.0** | 42.0 |
| **GPT-4o** | **75.7** | 75.3 | **59.0** | 55.0 | **75.7** | 75.0 | **59.0** | 58.0 |

Table 3: **Ablation Studies.** *Left:* Performance comparison between using engineered domain-specific verifier subsets (Eng) versus using all verifiers without tuning (All). *Right:* Performance comparison between using diverse verifiers from $\mathcal{M}^d$ (Diverse) versus querying the single best-performing verifier multiple times (Same). All metrics are accuracy (%).

## C  ANALYSIS: UNDERSTANDING MULTI-AGENT VERIFICATION

To better understand the key design choices that impact multi-agent verification, we conduct two ablation studies on MMLU-Pro and GPQA (diamond)—the two most challenging domains in our evaluation. We investigate: **(1)** how performance depends on engineering domain-specific sets of verifiers, and **(2)** whether using diverse verifiers outperforms repeatedly querying the best verifier.

**Effect of Verifier Engineering.** In Section 3.1, we introduced verifier engineering as an approach for selecting a relevant subset of verifiers $\mathcal{M}^d \subseteq \mathcal{M}$ for each domain $d$. Here, we compare our

engineered verifier subsets $\mathcal{M}^d$ against a simple baseline that uses all available aspect verifiers in $\mathcal{M}$ (see Appendix G.3 for a full list) without any domain-specific tuning. **??** (left) shows that engineering the set of verifiers is a more effective strategy. However, Table 5 in the Appendix shows that even the simple strategy of combining all verifiers in $\mathcal{M}$ remains competitive with both self-consistency and reward model verification baselines.

**Effect of Verifier Diversity.** Here, we investigate whether using diverse verifiers outperforms repeatedly querying a single verifier. Specifically, we compare the performance of our diverse domain-specific subsets $\mathcal{M}^d$ versus repeatedly querying the single best-performing verifier $v^* \in \mathcal{M}^d$ for domain $d$ (where the number of queries to $v^*$ equals $|\mathcal{M}^d|$). As shown in Table 3 (right), using diverse sets of verifiers generally outperforms querying the same verifier multiple times.

## D    SCALING UP TO 256 CANDIDATE OUTPUTS

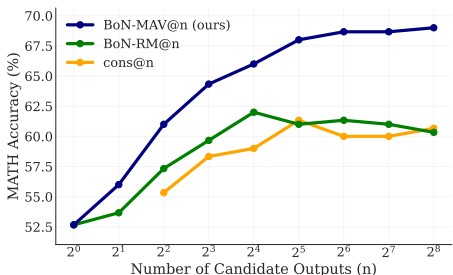 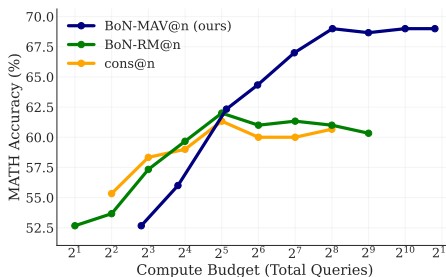

Figure 8: **Scaling to 256 candidate outputs.** Comparison of different test-time verification methods on MATH (Hendrycks et al., 2021) as we increase the number of candidates sampled from Gemini-1.5-Flash up to 256. *Left:* Accuracy (%) versus number of sampled outputs $n$. BoN-MAV consistently improves with additional samples while reward model verification (BoN-RM) and self-consistency (cons) plateau much earlier. *Right:* Accuracy (%) versus total compute budget (number of queries to both the generator and each verifier).

We extend our analysis to even larger scales by sampling 256 candidate outputs from Gemini-1.5-Flash on MATH. In Figure 8, we plot accuracy as a function of both the number of sampled candidate outputs $n$ (left) and the total compute budget (right). The left plot demonstrates that BoN-MAV consistently improves with additional samples, while reward model verification and self-consistency plateau early on. Starting at 52.7% base accuracy, the baselines plateau around 61% while BoN-MAV continues to 69%—nearly double the improvement. The right plot shows computational efficiency by comparing accuracy against the total compute budget, measured as the combined number of queries to both the generator and verifier models. At low compute budgets, the overhead of querying multiple verifiers with BoN-MAV means we can sample fewer candidate solutions, leading to initially worse performance than the baselines. However, once we have sufficient compute, BoN-MAV significantly outperforms both baseline methods.

## E    DISCUSSION

Multi-Agent Verification (MAV) introduces a promising dimension for scaling test-time compute: scaling the number of verifiers. In Section 3, we demonstrated that combining multiple verifiers enables more effective evaluation of candidate outputs, facilitates weak-to-strong generalization, and allows for self-improvement. However, our approach has important limitations and there are several opportunities for future work to explore.

First, our investigation is limited to a pool of 20 aspect verifiers based on just two base LLMs, and the design of our verifiers is constrained by our ability to come up with diverse verification strategies and relevant aspects. Future work could explore scaling to many more verifiers and try a more systematic exploration of the space of verifiers, potentially using LLMs themselves to generate diverse verification strategies and identify relevant aspects to verify. With better-engineered verifiers and more systematic exploration, we expect to observe stronger scaling patterns.

Second, our aggregation technique described in Section 2.2 uses a simple voting mechanism that directly sums the individual binary approvals from each verifier. This approach does not account for the confidence or relevance of each verifier, and verifiers do not observe each other's decisions or feedback. Future works could explore more sophisticated aggregation methods such as confidence-weighted voting or allowing verifiers to engage in debate (Du et al., 2023) before producing an approval. Moreover, our current approach uses a static engineered set of verifiers $\mathcal{M}^d$ for all questions in a domain $d$, even though it may be best to use fewer or different verifiers for specific questions. Future works could investigate dynamically selecting the best set of verifiers for particular problems or adaptively choosing additional verifiers based on the results of the first few verification queries. Additionally, the field of social choice theory (Arrow, 2012; Fishburn, 2015; Kelly, 2013; Brandt et al., 2016) is concerned with procedures for collective decision-making and might offer insights for aggregating the perspectives of diverse verifiers. Although, our setting differs in that we care more about verifier capabilities than preferences.

Next, our implementation of BoN-MAV is limited to only a single generator LLM. Thus, an interesting direction would be to explore sampling from multiple generators in addition to evaluating with multiple verifiers. Since different models may excel at solving different types of problems, this approach could make even better use of the growing ecosystem of LLMs and their diverse capabilities.

Furthermore, while our results show that BoN-MAV can improve language model performance at test-time, we did not investigate finetuning the generator LLM on the outputs selected by our verifiers. Similar to how prior works have finetuned on outputs selected through self-consistency (Huang et al., 2022) or reward models (Dong et al., 2023), training on outputs selected by MAV systems could be explored as a method to improve the generator LLM and also each of the LLM-based verifiers. Moreover, an interesting direction for future work is to directly use reinforcement learning to train both the generator and verifier models. That is, generator LLMs can be trained to maximize the scores across multiple verifiers, and the verifiers can simultaneously be trained to accurately verify individual aspects of responses.

Finally, multi-agent verification offers interesting opportunities for AI safety and oversight. The ability to combine multiple verifiers checking different aspects aligns with recent efforts towards safety checking the outputs of language models. That is, different verifiers can be engineered to check various safety and alignment properties, from basic constraints like avoiding harmful content to more nuanced properties like reasoning transparency. Our results on weak-to-strong generalization also align with recent work on scalable oversight, where weaker systems supervise stronger ones (Amodei et al., 2016; Saunders et al., 2022; Burns et al., 2023). In general, our work connects to broader ideas in AI alignment about using multiple models to improve safety (Irving et al., 2018).

An underlying thread throughout our work and discussion is the vision of a growing ecosystem of diverse language models that generate, verify, and learn from each other. Our work on multi-agent verification represents one step in this direction, and each of the future directions we have discussed offers a potential avenue for additional progress. We look forward to seeing how the research community advances these ideas.

## F    RELATED WORKS

**Scaling Test-Time Compute.**  Recent work has demonstrated that increasing computational resources during inference can significantly improve LLM performance (e.g., Wei et al. 2022; Snell et al. 2024). One line of research focuses on techniques where a single *generator* LLM produces additional output tokens during inference. These include scratchpads or Chain-of-Thought prompting (Nye et al., 2021; Wei et al., 2022), self-consistency or majority voting techniques (Wang et al., 2022; Li et al., 2022b; Thoppilan et al., 2022; Lewkowycz et al., 2022), and various self-reflection methods (e.g., Shinn et al. 2024; Qu et al. 2024; Madaan et al. 2024; Saunders et al. 2022; Bai et al. 2022b). Other works have explored training LLMs to generate special tokens which enhance reasoning ability at test-time (e.g., Goyal et al. 2023; Wang et al. 2023a; Herel & Mikolov 2024) or augmenting language models with tool-use abilities (e.g., Schick et al. 2023; Gao et al. 2023b; Qin et al. 2023; Qu et al. 2025).

Another line of research focuses on using a *verifier* model to evaluate the quality or correctness of outputs sampled from generator models (Cobbe et al., 2021; Zheng et al., 2023; Snell et al.,

2024). Typically, this is done through best-of-$n$ sampling (Stiennon et al., 2020; Cobbe et al., 2021; Nakano et al., 2021), where $n$ candidate outputs are generated and the highest-scoring output is selected based on some verifier. This verification can be performed at the outcome-level (Stiennon et al., 2020; Cobbe et al., 2021) or process-level (Lightman et al., 2023; Wang et al., 2024a). Recent works (Coste et al., 2023; Eisenstein et al., 2023) have also explored using ensembles of homogeneous reward models (identical model initializations trained on the same data but with different random seeds) to mitigate reward model overoptimization (Gao et al., 2023a). Additionally, some approaches allow reward models to produce their own Chain-of-Thought reasoning before scoring (Zhang et al., 2024b; Mahan et al., 2024). Various papers have combined language with search techniques at test-time, using verifiers to provide a heuristic signal. These verifiers may use LLMs as prompted value functions (e.g., Yu et al. 2023; Yao et al. 2024; Xie et al. 2024), incorporate real environment feedback (e.g., Zhou et al. 2023; Koh et al. 2024; Putta et al. 2024; Long 2023; Besta et al. 2024), or use trained value functions (e.g., Feng et al. 2023; Zhang et al. 2024a; Chen et al. 2024). Unlike prior works which typically rely on a single reward model verifier or homogeneous reward model ensembles trained on the same data, we propose a framework for combining multiple heterogeneous verifiers without additional training, and investigate scaling the number and type of verifiers as a novel test-time scaling dimension.

**Multi-Agent Reasoning with Language Models.** Recent works have investigated several approaches to multi-agent interaction for improving language model reasoning. Language model debate (e.g., Du et al. 2023; Chan et al. 2023; Pham et al. 2023; Liang et al. 2023; Subramaniam et al. 2025; Li et al. 2023; Cohen et al. 2023) and multi-agent discourse (e.g., Chen et al. 2023; Wang et al. 2023b; 2024b; Xu et al. 2023) have been studied as ways to enhance reasoning, and also as a direction for scalable oversight research (Irving et al., 2018). Prior works have also explored performing search with language models, which typically combines a generator LLM and a value model to guide exploration (see the previous paragraph). Moreover, some works have explored multi-modal reasoning through agent collaboration (e.g., Zeng et al. 2022; Li et al. 2022a; Ajay et al. 2023; Jiang et al. 2024). Unlike prior work on multi-agent reasoning which focuses on collaborative problem-solving, we introduce a framework specifically for scaling test-time verification by combining multiple verifiers without training.

## G  EXPERIMENTAL SETUP

### G.1  DOMAINS

We evaluate our method on the following four domains:

- **Mathematics.** The MATH dataset (Hendrycks et al., 2021) consists of competition-level math questions at five difficulty levels. For our experiments, we randomly sample 400 questions from the test set across all five levels: 100 for validation and 300 for testing.
- **General Knowledge & Reasoning.** MMLU-Pro (Wang et al., 2024c) is an enhanced version of the popular MMLU benchmark (Hendrycks et al., 2020) which features more challenging, reasoning-focused questions and expands the multiple-choice set from four to ten options. As with MATH, we sample 100 questions for validation and 300 for testing.
- **Graduate-Level Reasoning.** The GPQA dataset (Rein et al., 2023) consists of graduate-level, multiple-choice questions in biology, physics, and chemistry. For our experiments, we utilize GPQA's "diamond" subset — a collection of 198 high-quality and extremely challenging questions. We sample 98 questions for validation and 100 for testing.
- **Coding.** HumanEval (Chen et al., 2021) is a widely-used benchmark consisting of 164 Python programming questions. We sample 64 questions for validation and 100 for testing.

### G.2  VERIFIER ENGINEERING

For our experiments, we implement the verifier engineering method described in Section 2.4. To create our initial diverse pool $\mathcal{M}$ of 20 aspect verifiers, we vary the three key axes that define aspect verifiers: (1) Base model: Gemini-1.5-Flash or GPT-4o-mini, (2) Aspect to verify: Mathematical correctness, logical soundness, factuality, etc., and (3) Verification strategy: Direct approval, step-by-step verification, solution rephrasing, edge case checking, etc. From this pool, we then select domain-specific subsets $\mathcal{M}^d \subseteq \mathcal{M}$ that maximize average performance across all generator LLMs

on the corresponding validation sets. The complete list of verifiers and the domain-specific subsets are detailed in Table 4. We choose Gemini-1.5-Flash and GPT-4o-mini as the base LLMs for our aspect verifiers since they are cost-effective for large-scale verification and enable us to demonstrate that combining multiple weaker verifiers can improve the performance of even stronger generator LLMs (Section 3.2).

### G.3 ASPECT VERIFIER SUBSETS

Table 4 outlines all 20 aspect verifiers in $\mathcal{M}$ and which ones were selected for each domain-specific subset $\mathcal{M}^d$.

| Base Model | Aspect to Verify | Verification Strategy | MATH | MMLU-Pro | GPQA | HumanEval |
|---|---|---|:---:|:---:|:---:|:---:|
| GPT-4o-mini | Math Correctness | Step-by-Step | | ✓ | ✓ | ✓ |
| | Logical Soundness | Step-by-Step | | ✓ | ✓ | ✓ |
| | Factual Correctness | Step-by-Step | | | | ✓ |
| | Unit Conversions | Step-by-Step | ✓ | | ✓ | ✓ |
| | General Correctness | Direct Approval | | | | ✓ |
| | General Correctness | Summarize Solution | ✓ | | | |
| | General Correctness | Explain Differently | | ✓ | ✓ | ✓ |
| | General Correctness | Edge Cases | ✓ | ✓ | | ✓ |
| | General Correctness | Common Mistakes | ✓ | ✓ | | |
| | General Correctness | Domain Knowledge | ✓ | ✓ | | ✓ |
| Gemini-1.5-Flash | Math Correctness | Step-by-Step | | | | |
| | Logical Soundness | Step-by-Step | | | | ✓ |
| | Factual Correctness | Step-by-Step | | | | |
| | Unit Conversions | Step-by-Step | | ✓ | ✓ | ✓ |
| | General Correctness | Direct Approval | | | | ✓ |
| | General Correctness | Summarize Solution | | | | ✓ |
| | General Correctness | Explain Differently | | | ✓ | ✓ |
| | General Correctness | Edge Cases | ✓ | | | |
| | General Correctness | Common Mistakes | | ✓ | ✓ | |
| | General Correctness | Domain Knowledge | | | | ✓ |
| **Total Verifiers Used** | | | **6** | **8** | **7** | **14** |

Table 4: Overview of all aspect verifiers in $\mathcal{M}$. Checkmarks (✓) indicate which verifiers were selected for each domain-specific subset $\mathcal{M}^d$. The table shows all 20 combinations of base models, aspects to verify, and verification strategies that we created (10 per base model). The bottom row shows the number of verifiers $|\mathcal{M}^d|$ for each domain.

### G.4 GENERATOR LLMS

We evaluate eight generator LLMs (four closed-source models and four open-source models) and restrict our set of generator models to those released before September 2024. For closed-source models, we use gemini-1.5-flash-001 and gemini-1.5-pro-001 (Team et al., 2024a), as well as gpt-4o-mini-2024-07-18 and gpt-4o-2024-08-06 (Achiam et al., 2023). For open-source models, we use Mistral-7B-v0.3 (Jiang et al., 2023), Llama-3.1-8B (Dubey et al., 2024), Gemma-2-9B, and Gemma-2-27B (Team et al., 2024b).

## G.5 Reward Model Baseline

Our reward model verification baseline (BoN-RM) uses `Skywork/Skywork-Reward-Llama-3.1-8B-v0.2` (Liu et al., 2024), the top scoring open-source 8B reward model on RewardBench (Lambert et al., 2024) at the time of writing. This pretrained reward model outperforms numerous larger models including 70B and 340B models, and can be run on academic-scale compute.

## G.6 Prompts

For generator LLMs, we use a consistent prompt format across all models while varying the content by domain. Table 6 contains these domain-specific prompts.

For aspect verifiers, each prompt consists of two components:

1. A domain-dependent system prompt (Table 7) that establishes the verification context (e.g., mathematical problems, multiple-choice questions, or code implementations)
2. A domain-independent verification prompt (Table 8 and Table 9) that specifies the aspect to verify and verification strategy

This two-part structure allows us to combine any aspect-strategy verification method with any domain while maintaining consistent evaluation criteria across base models.

## H Additional Results

Table 5 compares BoN-MAV using all 20 aspect verifiers in $\mathcal{M}$ (without domain-specific engineering) against self-consistency and reward model verification. Even without engineering domain-specific subsets $\mathcal{M}^d$, combining all verifiers remains competitive with baseline methods.

| | MMLU-Pro | | | | GPQA (diamond) | | | |
|---|---|---|---|---|---|---|---|---|
| **Generator Model** | MAV-All | Cons | RM | pass@1 | MAV-All | Cons | RM | pass@1 |
| **Gemini-1.5-Flash** | **65.7** | 63.3 | 60.7 | 59.3 | 41.0 | 40.0 | **46.0** | 42.0 |
| **Gemini-1.5-Pro** | 70.3 | **71.7** | 69.3 | 68.0 | **49.0** | 45.0 | **49.0** | 45.0 |
| **GPT-4o-mini** | **65.3** | 63.7 | 62.7 | 62.3 | **49.0** | 48.0 | 44.0 | 38.0 |
| **GPT-4o** | 75.3 | **76.3** | 72.7 | 73.3 | 55.0 | **59.0** | 58.0 | 54.0 |

Table 5: Performance (accuracy %) of BoN-MAV with all 20 aspect verifiers (without any tuning, labeled as MAV-all in the table) compared to reward model verification (RM), self-consistency (Cons), and the base pass@1 accuracy of the generator LLM. Using all verifiers without domain-specific tuning remains competitive with reward model verification and self-consistency.

## I Additional Illustrations

Figure 9, Figure 10, and Figure 11 provide additional examples of how multiple aspect verifiers evaluate a single candidate output. Figure 9 demonstrates verification using multiple strategies with a single base model on MATH (Hendrycks et al., 2021). Figure 10 shows verification of a coding solution from HumanEval (Chen et al., 2021). Figure 11 illustrates verification of a correct solution from GPQA (diamond) (Rein et al., 2023), showing how different base models can assess the same aspect differently. Each figure follows the same format as Figure 5 from the main paper.

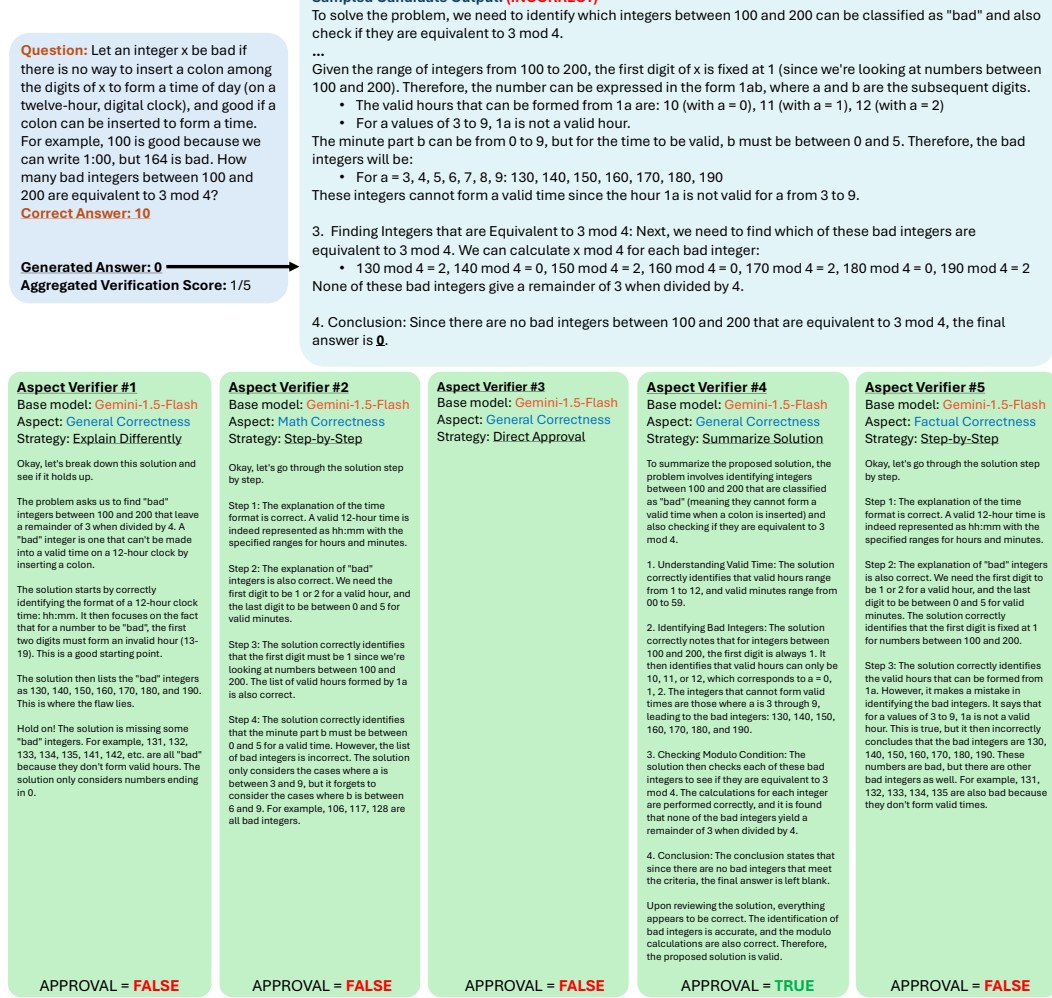

Figure 9: **Multi-agent verification for a single solution (additional example).** An illustration similar to Figure 5. Five different aspect verifiers evaluate an incorrect MATH (Hendrycks et al., 2021) solution sampled from Gemini-1.5-Pro. All verifiers use Gemini-1.5-Flash as the base model but vary in their aspects to verify (e.g., general correctness, mathematical correctness) and verification strategies (e.g., direct approval, step-by-step verification). Four verifiers correctly identify the error, while one verifier using general correctness through summarization incorrectly approves the solution. This demonstrates how diverse verification methods can produce more reliable signals even when using a single base model, as multiple verifiers can compensate when another fails.

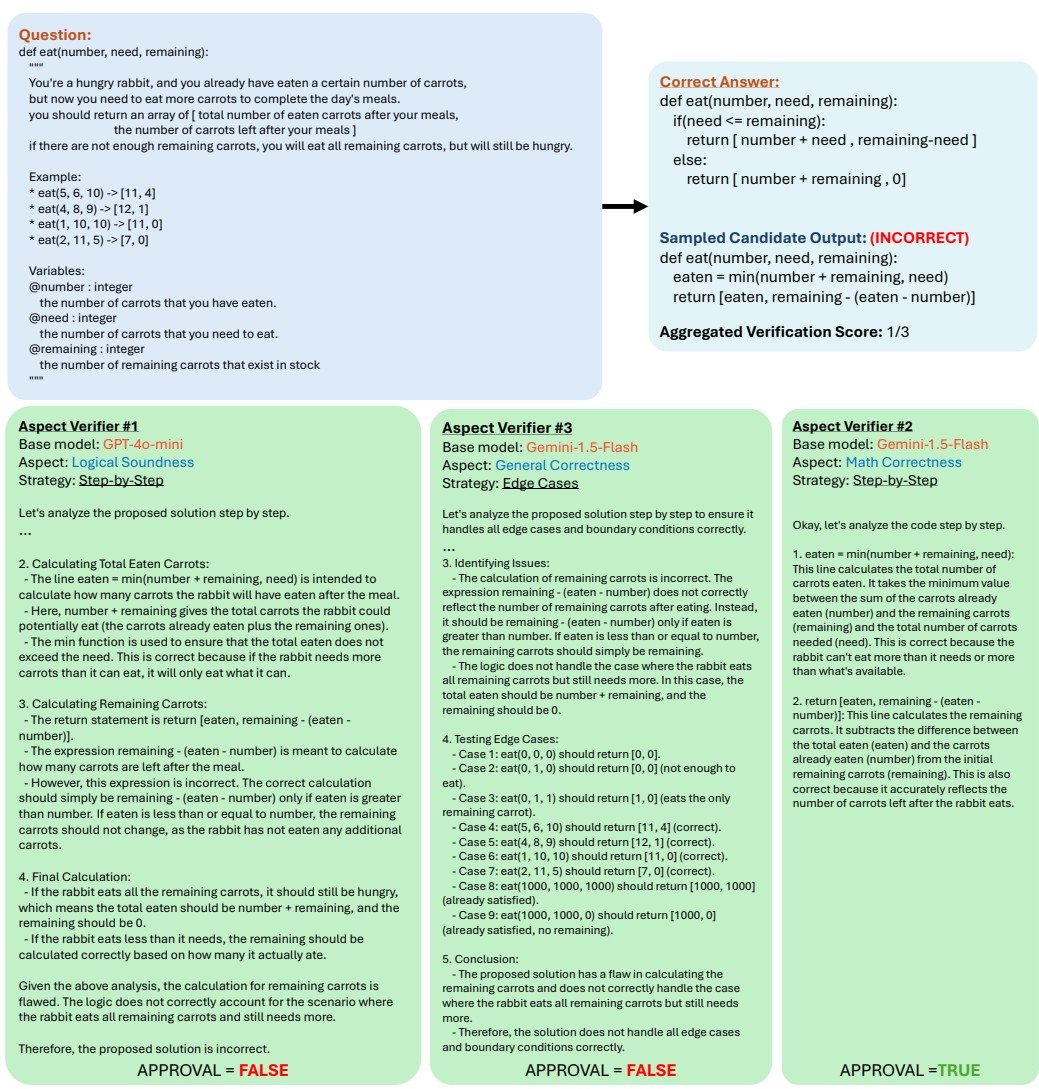

Figure 10: **Multi-agent verification for a single solution (additional example).** An illustration similar to Figure 5. Here, three different aspect verifiers evaluate an incorrect HumanEval (Chen et al., 2021) solution sampled from Gemini-1.5-Flash. Two verifiers correctly identify the error through careful analysis, while one verifier incorrectly approves the solution.

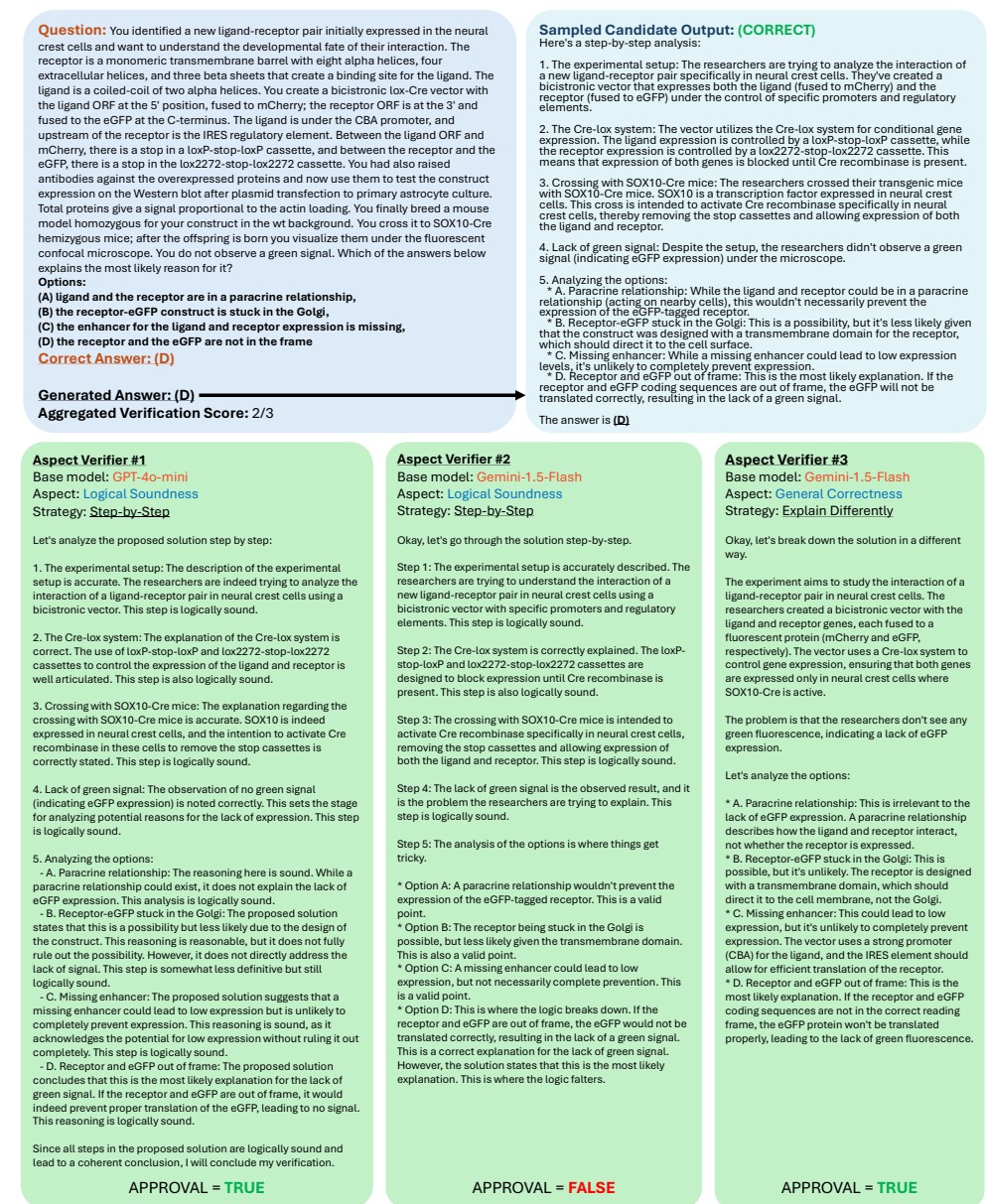

Figure 11: **Multi-agent verification for a single solution (additional example).** An illustration similar to Figure 5. Here, three different aspect verifiers evaluate a correct GPQA (diamond) (Rein et al., 2023) solution sampled from Gemma-2-27B. The verifiers vary in their base models, aspects to verify, and verification strategies. Notice that Gemini-1.5-Flash incorrectly rejects the solution when evaluating logical soundness but correctly approves it when prompted to explain the solution differently. Meanwhile, GPT-4o-mini correctly approves the solution when evaluating logical soundness. Different base models can produce different evaluations of the same aspect.

| Domain | Generator Prompt |
|---|---|
| MATH | *You are a helpful assistant skilled in math problem-solving. Always end your solution with the final numerical answer enclosed in LaTeX \boxed{} notation. If there is no solution, reply with an empty \boxed{}. Please solve the following math problem step by step: < Question > Provide your detailed solution below:* |
| MMLU-Pro | *Answer the following multiple choice question. Think step by step before answering, and then output the answer in the format of "The answer is (X)" at the end, where X is the LETTER of the correct answer.*
*QUESTION: < Question >*
*Think step by step, then end with EXACTLY "The answer is (X)", where X is the LETTER of the correct answer. Do not include the answer text itself, only the letter.* |
| GPQA (diamond) | Same as MMLU-Pro. |
| HumanEval | *Read the following function signature and docstring, and fully implement the function described. Your response should only contain the code for this function.*
*< Function Signature and Docstring >* |

Table 6: **Generator Prompts.** Generator prompts by domain. Each domain uses one consistent prompt across all generator LLMs.

| Domain | Aspect Verifier System Prompt |
|---|---|
| MATH | *You are a critical verifier tasked with evaluating mathematical problem-solving. You will be presented with a question and a proposed solution. Your job is to carefully go over and analyze the solution. Follow the instructions.* |
| MMLU-Pro | *You are a critical verifier tasked with evaluating multiple-choice question-answering. You will be presented with a question, the multiple-choice options, and a proposed solution. Your job is to carefully go over and analyze the solution. Follow the instructions.* |
| GPQA (diamond) | Same as MMLU-Pro. |
| HumanEval | *You are a critical verifier tasked with evaluating code implementations. You will be presented with a prompt and a code implementation. Your job is to carefully go over and analyze the code. Follow the instructions.* |

Table 7: **Aspect Verifier System Prompts.** System prompts for aspect verifiers. These provide domain-specific context for the verification instructions in Table 8 and Table 9.

| Aspect to Verify | Verification Strategy | Aspect Verifier Prompt |
|---|---|---|
| Mathematical Correctness | Step-by-Step | *QUESTION: <Question>*
*PROPOSED SOLUTION: <Solution>*
*INSTRUCTIONS: Go over each step in the proposed solution and check whether it is mathematically correct. Think out load. If you reach a step that is incorrect, stop and reply 'FINAL VERIFICATION ANSWER: False'. If you get to the end of all the steps and each step was correct, reply 'FINAL VERIFICATION ANSWER: True'.* |
| Logical Soundness | Step-by-Step | *QUESTION: <Question>*
*PROPOSED SOLUTION: <Solution>*
*INSTRUCTIONS: Go over each step in the proposed solution and check whether it is logically sound. Think out load. If you reach a step that is not logically sound, stop and reply 'FINAL VERIFICATION ANSWER: False'. If you get to the end of all the steps and each step was logically sound, reply 'FINAL VERIFICATION ANSWER: True'.* |
| Factual Correctness | Step-by-Step | *QUESTION: <Question>*
*PROPOSED SOLUTION: <Solution>*
*INSTRUCTIONS: Go over each step in the proposed solution and check whether the facts presented are correct. Think out load. If you reach a step with incorrect facts, stop and reply 'FINAL VERIFICATION ANSWER: False'. If you get to the end of all the steps and each step had correct facts, reply 'FINAL VERIFICATION ANSWER: True'.* |
| Unit Conversions | Step-by-Step | *QUESTION: <Question>*
*PROPOSED SOLUTION: <Solution>*
*INSTRUCTIONS: Check if the units are handled correctly in each step of the solution. Think out loud. If you find any issues with the units, stop and reply 'FINAL VERIFICATION ANSWER: False'. If all units are handled correctly, reply 'FINAL VERIFICATION ANSWER: True'.* |

Table 8: **Aspect Verifier Prompts (Part 1).** Aspect verifier prompts for each aspect-strategy combination. These prompts follow the system prompts in Table 7.

| Aspect to Verify | Verification Strategy | Aspect Verifier Prompt |
|---|---|---|
| General Correctness | Direct Approval | *QUESTION: <Question>*
*PROPOSED SOLUTION: <Solution>*
*INSTRUCTIONS: Is this solution correct for the given question? Respond with ONLY 'FINAL VERIFICATION ANSWER: True' or ONLY 'FINAL VERIFICATION ANSWER: False'. Do not provide any explanation or additional text.* |
| General Correctness | Summarize Solution | *QUESTION: <Question>*
*PROPOSED SOLUTION: <Solution>*
*INSTRUCTIONS: Summarize the solution in your own words, explore anything you think may be incorrect. Think out load. If you find something that's incorrect, stop and reply 'FINAL VERIFICATION ANSWER: False'. If you've gone over the solution and everything seems correct, reply 'FINAL VERIFICATION ANSWER: True'.* |
| General Correctness | Explain Differently | *QUESTION: <Question>*
*PROPOSED SOLUTION: <Solution>*
*INSTRUCTIONS: Explain the solution in a different way than it was presented. Try to find any flaws in the solution. Think out load. If you find something that's incorrect, stop and reply 'FINAL VERIFICATION ANSWER: False'. If you've gone over the solution and everything seems correct, reply 'FINAL VERIFICATION ANSWER: True'.* |
| General Correctness | Edge Cases | *QUESTION: <Question>*
*PROPOSED SOLUTION: <Solution>*
*INSTRUCTIONS: Check if the solution handles edge cases and boundary conditions, test extreme values or special cases. Think out loud. If any boundary conditions or edge cases fail, stop and reply 'FINAL VERIFICATION ANSWER: False'. If all boundary conditions and edge cases are handled correctly, reply 'FINAL VERIFICATION ANSWER: True'.* |
| General Correctness | Common Mistakes | *QUESTION: <Question>*
*PROPOSED SOLUTION: <Solution>*
*INSTRUCTIONS: Check if the solution has any common mistakes, calculation errors, or misconceptions that typically found in this type of problem. Think out loud. If you find any common mistakes, stop and reply 'FINAL VERIFICATION ANSWER: False'. If no common mistakes are found, reply 'FINAL VERIFICATION ANSWER: True'.* |
| General Correctness | Domain Knowledge | *QUESTION: <Question>*
*PROPOSED SOLUTION: <Solution>*
*INSTRUCTIONS: Check if the solution correctly applies relevant domain-knowledge, established theories, and standard practices for this type of problem. Think out loud. If any domain knowledge is misapplied or violated, stop and reply 'FINAL VERIFICATION ANSWER: False'. If all domain-specific knowledge is correctly applied, reply 'FINAL VERIFICATION ANSWER: True'.* |

Table 9: **Aspect Verifier Prompts (Part 2).** Aspect verifier prompts for each aspect-strategy combination. These prompts follow the system prompts in Table 7.

