# OpenReview forum: "Multi-Agent Verification: Scaling Test-Time Compute with Multiple Verifiers (Abridged)"
_ICLR.cc/2025/Workshop/MCDC — MCDC @ ICLR 2025_

### Official Review · Reviewer_KJJr · 2025-02-23

**Rating:** 6
**Confidence:** 4
**Fit:** 4

**Summary:**

This work introduces Goal Verifiers (GVs), external large language models (LLMs) that assess the correctness of a solution from different perspectives at test time. GVs do not require additional training and naturally integrate multiple verification signals through simple voting mechanisms. Additionally, this work proposes Multi-Agent Verification (MAV), which leverages multiple GVs to enhance verification reliability. The results demonstrate that model performance improves as both the number and diversity of GVs increase, highlighting the effectiveness of MAV in refining LLM outputs.

**Reason For Giving A Higher Score:**

See strengths.

**Reason For Giving A Lower Score:**

See weakness.

**Strengths And Weaknesses:**

### **Strengths**
1. The approach of scaling test-time computation by incorporating more verifiers is intuitive. Experimental results confirm that it outperforms self-consistency and reward model verification.
2. The verifier requires no additional training and can evaluate solutions from different verification aspects, making it a flexible and adaptable approach.

### **Weaknesses**
1. The performance **highly depends** on the strength of the verifiers. In this paper, the authors use strong closed-source models, Gemini-1.5-Flash and GPT-4o-mini. The overall performance may degrade when using smaller or weaker models as verifiers.
2. Although the verifier does not require additional training, the need for multiple verifiers to cover diverse aspects of verification could still introduce significant computational costs.

**Suggestions:**

1. Consider including experimental results using weaker models as verifiers to analyze performance degradation and provide insights into the robustness of the proposed approach.
2. It may be useful to explore more stable verification methods, such as static verifiers, to reduce reliance on LLM-based verification. However, static verifiers might lack the flexibility that LLM-based verifiers provide.

---

### Official Review · Reviewer_vKyS · 2025-02-27

**Rating:** 5
**Confidence:** 2
**Fit:** 3

**Summary:**

The paper proposes to verify MATH solutions by an ensemble of 'multi-agent' verifiers (MAV): the algorithms goes as:
1) generate different MATH solutions for a problem;
2) prompt different models (gpt-4o-mini, gemini-flash) with different prompts to predict the quality of each solution;
3) aggregate the verifications;
4) rerank the solutions.
MAV performs better than consistency or RM scoring. The paper is mildly in topic for this workshop, the closest aspect being the usage of different models to verify the solutions (which might be different in architecture from the original generators).

**Reason For Giving A Higher Score:**

n/a

**Reason For Giving A Lower Score:**

n/a

**Strengths And Weaknesses:**

\+ good perf on using ensemble of verifiers

\+ nicely written

\- mildly in topic for workshop

\- couldn't find analysis about diversity of the verifiers

\- comparison btw using different models vs different GVs for the same model

**Suggestions:**

Reason for low-ish score is workshop topic fit. I think it would be more interesting for this workshop if the paper focused on analyzing the diversity of the verifiers and how perf changes w. the diversity of the verifiers. the abstract says : "where performance improves with both the number and diversity of GVs", it could be my misunderstanding but I couldn't find an analysis of the diversity of the verifiers and how this impact the performance, which might be more interesting for this workshop.

Paper seems to be missing a baseline which is basically using gemini or gpt-4o-mini only across GVs?

Using open weights model, rather than gemini or gpt-4o-mini, might also be interesting. Which models should we choose? Is there a way to get to know which model would be better for this task if I have a set of them available?

---

### Official Review · Reviewer_9y4h · 2025-02-27

**Rating:** 6
**Confidence:** 4
**Fit:** 4

**Summary:**

This work studies the important problem of improving test-time performance by introducing Goal Verifiers that are LMs prompted to provide binary scores for a generated response. By scaling the number of Goal Verifiers, the generated responses can be judged on several dimensions. A Best of N Multi-Agent Verification mechanism is proposed that samples n responses from the generator and runs them against the Goal Verifiers. Then the solution receiving the highest total score from the Verifiers is chosen as the final response. Experiments are run on datasets like MATH, HumanEval, GPQA, MMLU with 5 different LMs to demonstrate the effectiveness of MAV.

**Reason For Giving A Higher Score:**

N/A

**Reason For Giving A Lower Score:**

Refer weakness

**Strengths And Weaknesses:**

Strengths
- Well-written paper, easy to follow
- Several experiments across domains and LMs

Weakness
- The idea is itself not new, but has been presented well. Similar solutions of sampling n responses and choosing the best based on some form of critique mechanism has already been well explored in the literature

**Suggestions:**

- "combining multiple relatively weak verifiers together can enhance the performance of stronger generator models" I found this interesting. It would be nice to have some analysis on the cost effectiveness of this approach compared to baselines?
- An interesting aspect would be leveraging the multi-agent framework to more effectively score the generations via Debate style methods.

---

### Decision · Program_Chairs · 2025-03-06

**Decision:**

Accept

**Comment:**

This paper proposes to use multiple external verifiers to assess the solution quality of a model and provide a better verification signals through simple voting mechanisms. This is a novel and timely idea that shows some potential. We encourage the authors to take reviewers' comments and suggestions into consideration for the final version of the paper.